# Axonal growth on surfaces with periodic geometrical patterns

**Jacob P. Sunnerberg**[1], **Marc Descoteaux**[1], **David L. Kaplan**[2], **Cristian Staii**[1] *

**1** Department of Physics and Astronomy, Tufts University, Medford, Massachusetts, United States of America, **2** Department of Biomedical Engineering, Tufts University, Medford, Massachusetts, United States of America

* cristian.staii@tufts.edu

**Data Availability Statement:** All relevant data are within the manuscript and its Supporting Information files.

**Funding:** CS, National Science Foundation award DMR 2104294.

## Abstract

The formation of neuron networks is a complex phenomenon of fundamental importance for understanding the development of the nervous system, and for creating novel bioinspired materials for tissue engineering and neuronal repair. The basic process underlying the network formation is axonal growth, a process involving the extension of axons from the cell body towards target neurons. Axonal growth is guided by environmental stimuli that include intercellular interactions, biochemical cues, and the mechanical and geometrical features of the growth substrate. The dynamics of the growing axon and its biomechanical interactions with the growing substrate remains poorly understood. In this paper, we develop a model of axonal motility which incorporates mechanical interactions between the axon and the growth substrate. We combine experimental data with theoretical analysis to measure the parameters that describe axonal growth on surfaces with micropatterned periodic geometrical features: diffusion (cell motility) coefficients, speed and angular distributions, and axon bending rigidities. Experiments performed on neurons treated Taxol (inhibitor of microtubule dynamics) and Blebbistatin (disruptor of actin filaments) show that the dynamics of the cytoskeleton plays a critical role in the axon steering mechanism. Our results demonstrate that axons follow geometrical patterns through a contact-guidance mechanism, in which high-curvature geometrical features impart high traction forces to the growth cone. These results have important implications for our fundamental understanding of axonal growth as well as for bioengineering novel substrates that promote neuronal growth and nerve repair.

## Introduction

Neurons are the basic cells of the nervous system. During their growth neurons extend two types of processes: axons and dendrites, which navigate to other neurons and form complex neuronal networks that transmit electrical signals throughout the body. The extension of the axon is guided by its growth cone, a motile unit located at the distal tip of the axon that navigates through the surrounding environment using electrical, chemical, mechanical, and morphological cues [1–4]. The dynamics of the growth cone is controlled by a flexible ensemble of actin and microtubule filaments that form the neuron cytoskeleton [1–7].

**Competing interests:** The authors have declared that no competing interests exist.

Previous research has identified many of the molecular pathways responsible for intercellular signaling in the formation of neuronal networks [1–7]. The biomechanical properties of neurons are an integral part of their functional behavior and play an essential role in the normal brain development. For example, it is known that the growing axon is capable of detecting a large variety of biochemical, mechanical and topographical cues within the growth environment, and of directing its growth over relatively long distances (hundreds of microns) with great precision [1–8]. To understand how neurons grow axons and dendrites and wire up the nervous system, we need to understand how they respond to external physical stimuli.

Much of the research into how geometric and mechanical cues affect neuronal growth has been performed *in vitro* on ensembles of neurons grown on substrates where the geometry can be controlled. These studies have shown that neurons grown on substrates with periodic geometrical features develop different growth patterns when compared to neurons grown on surfaces lacking a periodic geometry. Such differences include populations of axons which are significantly longer and which tend to align their growth along preferred spatial directions [9–16].

The ability to control and direct neuronal growth *in vitro* has important consequences for exploiting biologically inspired designs for applications in tissue engineering, neural repair, and *in vitro-in vivo* device interfaces. A major goal in neural tissue engineering is to create controlled environments that promote axonal growth and reproduce the physiological conditions found *in vivo* [3,5,9–11,15,17,18]. However, there are still major challenges with respect to the ability to control and direct neuronal growth. For example, despite recent advances, there are still key unanswered questions about the mechanisms that control neuron biomechanical responses, as well as about the details of cell-substrate interactions such as the synergy or antagonism between various external cues. Furthermore, most of the previous work on studying neuronal growth *in vitro* has focused on qualitative or semi-quantitative models to describe the influence of geometrical or mechanical cues on the formation of neuronal network. A detailed characterization of the basic mechanisms that underlie the growth cone response to physical cues is still missing.

In our previous work we have shown that axonal growth on surfaces with controlled geometries arises as the result of an interplay between deterministic and stochastic components of growth cone motility [10,15,16,19,20]. Deterministic influences include, for example the presence of preferred directions of growth along specific geometric patterns on substrates, while stochastic components come from the effects of polymerization of cytoskeletal elements (actin filaments and microtubules), neuron signaling, low concentration biomolecule detection, biochemical reactions within the neuron, and the formation of lamellipodia and filopodia [1,2,7,21]. The resultant growth cannot be predicted for individual neurons due to this stochastic-deterministic interplay, however the growth dynamics for a population of neurons can be modeled by probability functions that satisfy a set of well-defined stochastic differential equations, such as Langevin and Fokker-Planck equations [10,15,19,20,22–29]. In previous work we have shown that axonal dynamics on uniform glass surfaces is described by an Ornstein-Uhlenbeck (Brownian) process, defined by a linear Langevin equation and stochastic white noise [19,20]. We have also reported that neurons cultured on poly-D-lysine coated polydimethylsiloxane (PDMS) substrates with periodic parallel ridge micropatterns of spatial periodicity $d$ (henceforth referred to as the pattern spatial period), grow axons parallel to the surface patterns. We have studied axonal growth as a function of time on these micropatterned surfaces and found that axonal alignment increases as a function of time [16]. The axonal dynamics is described by non-linear Langevin equations, involving quadratic velocity terms and non-zero coefficients for the angular orientation of the growing axon [20]. In another paper we have used the Langevin and Fokker-Planck equations to quantify axonal growth on

surfaces with ratchet-like topography (asymmetric tilted nanorod: nano-ppx surfaces) [10]. We have shown that the axonal growth is aligned with a preferred spatial direction as a result of a "deterministic torque" that drives the axons to directions determined by the substrate geometry. We have also measured the angular distributions and the coefficients of diffusion and angular drift on these substrates [10]. Our results provide a detailed analysis of axonal growth on substrates with different geometrical patterns by measuring speed and acceleration distributions as a function of substrate geometry [20], axonal alignment as a function of time [16], as well as axonal angular distributions, angular drift and diffusion coefficients [10,16,20].

In this paper we show that axonal dynamics on micropatterned surfaces is controlled by a contact-guidance mechanism, which stems from the coupling between the axon and the high-curvature geometrical features of the growth substrate. We develop a quantitative model of growth cone motility which incorporates cellular stiffness and the neuron ability to sense the substrate geometry. This theoretical model fully accounts for the experimental data measured on ensembles of axons, including speed distributions and angular alignment. Furthermore, we perform experiments which demonstrate that inhibition of cytoskeletal dynamics by treatment of neurons with Taxol (inhibitor of microtubules) and Blebbistatin (inhibitor of myosin II and actin dynamics) results in a significant decrease of the axonal alignment, by altering the cell-substrate interactions. Our results show that the motion of axons on surfaces with micropatterned periodic patterns is governed by the interplay between axonal stiffness and substrate geometry that leads to axonal alignment on these surfaces. This work provides new insights for creating bioinspired systems that emulate neuronal growth *in vivo*, and should have significant impact for designing new platforms for guiding growth and regeneration of neurons.

## Materials and methods

### Surface preparation and AFM imaging

The periodic micropatterns on PDMS surfaces consist of parallel ridges separated by troughs. Each surface is characterized by two different geometrical parameters: 1) the value of the pattern spatial period *d*, defined as the distance between two neighboring ridges (Fig 1A), and 2) the radius of curvature *R* of the semi-cylindrical pattern (Fig 1B). These parameters were measured for each type of surface using the Atomic Force Microscope (AFM) (Fig 1A). All PDMS patterns have been imaged using an MFP3D AFM (Asylum Research/Oxford Instruments). The AFM topographical images of the surfaces were obtained using the AC mode of operation, and AC 160TS cantilevers (Asylum Research, Santa Barbara, CA).

To prepare the periodic geometrical patterns we used a simple fabrication method based on imprinting diffraction grids with different grating constants onto PDMS substrates. We started with 20mL polydimethylsiloxane (PDMS) solution (Silgard, Dow Corning) and pour it over diffraction gratings with slit separations: 1 μm—6 μm (in increments of 1 μm) and total surface area 25 x 25 mm$^2$ (Scientrific Pty. and Newport Corp. Irvine, CA). The PDMS films were left to polymerize for 48 hrs at room temperature, then peeled away from the diffraction gratings and cured at 55$^0$ C for 3 hrs. We used AFM imaging to ensure that the pattern was successfully transferred from the diffraction grating to the PDMS surface (Fig 1A). The result is a series of periodic patterns (parallel lines with crests and troughs) with constant distance *d* between two adjacent lines, and constant curvature radius *R* for each type of surface. By using different diffraction gratings we obtained different growth surfaces with spatial periods in the range *d* = 1 μm to 6 μm (in increments of 1 μm), which correspond to curvature radii *R* = 0.5 μm to 1 μm (in increments of 0.1 μm). Fig 1A shows an example of an AFM image of PDMS micropatterns with *d* = 5 μm and *R* = 0.9 μm. The AFM data demonstrates that the patterns are

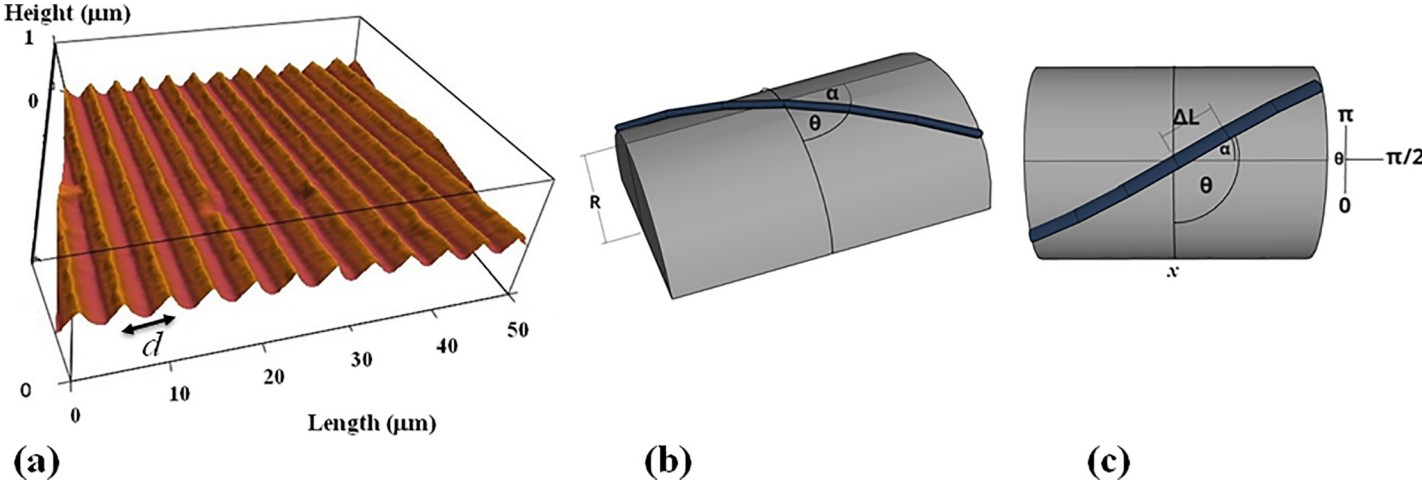

**Fig 1.** (a) Example of an Atomic Force Microscope (AFM) topographic image of a PDL coated PDMS patterned surface. The image shows that the micropatterns are periodic in the $x$ direction (defined as the direction perpendicular to the patterns). In this image the patterns have a spatial period $d = 5$ µm and a constant profile shape with a depth of approximately 0.6 µm. (b) Schematic of an axon growing on the geometrical pattern. The schematic shows the coordinate system and the definition of the angular coordinate $\alpha$. For each axonal segment of length $\Delta L$ (defined in the main text) the growth angle $\alpha$ is defined as the angle that the axon is making with the long axis of the geometrical PDMS pattern. The angular coordinate $\theta$ used for the data analysis is defined as the complementary angle: $\theta = \frac{\pi}{2} \pm \alpha$.

periodic and have constant depth. Control AFM experiments demonstrated that the topographical and mechanical properties of the micropatterned PDMS substrates did not change significantly upon coating with PDL or among surfaces with different spatial periods (S1 and S2 Figs). The surfaces were then glued to glass slides using silicone glue and dried for 48 hours. Next, each surface was cleaned with sterile water and spin-coated with 3 mL of Poly-D-lysine (PDL) (Sigma-Aldrich, St. Louis, MO) solution of concentration 0.1 mg/mL. The spinning was performed for 10 minutes at 1000 RPM. Prior to cell culture the surfaces have been sterilized using ultraviolet light for 30 minutes.

## Cell culture

The cells used in this work were cortical neurons obtained from embryonic day 18 rats. For cell dissociation and culture we have used established protocols presented in our previous work [10,15,16,19,20,30–33]. The brain tissue protocol was approved by Tufts University Institutional Animal Care Use Committee and complies with the NIH guide for the Care and Use of Laboratory Animals. The cortices have been incubated in 5 mL of trypsin at 37˚C for 20 minutes. To inhibit the trypsin we have used 10 mL of soybean trypsin inhibitor (Life Technologies). Next, the neuronal cells have been mechanically dissociated, centrifuged, and the supernatant was removed. After this step the neurons have been re-suspended in 20 mL of neurobasal medium (Life Technologies) enhanced with GlutaMAX, b27 (Life Technologies), and pen/strep. Finally, the neurons have been re-dispersed with a pipette, counted, and plated on micropatterned polydimethylsiloxane (PDMS) substrates coated with poly-D-lysine (PDL), at a density of 4,000 cells/cm$^2$. We have previously reported that neurons cultured at relatively low densities (in the range 3000 to 7000 cells/cm$^2$) were optimal for studying axonal growth on surfaces with different mechanical, geometrical and biochemical properties [10,15,16,19,20,30–32]. These densities are also relevant for growing axons in the early stages of development of the nervous system [1,2]. In our previous work we have performed immunostaining experiments which show high neuron cell purity in these cultures [30].

## Fluorescence imaging

The MFP3D AFM is equipped with a BioHeater closed fluid cell, and an inverted optical microscope (Micro Video Instruments, Avon, MA). All neuronal cells were imaged using fluorescence microscopy on the inverted optical stage (Nikon Eclipse Ti) of the AFM. We have previously shown that both untreated and chemically modified neurons grown in the MFP3D fluid cell remain viable over long periods of time [10,15,16,19,20,30–32]. For fluorescence imaging the cortical neurons cultured on glass or PDMS surfaces, were rinsed with phosphate buffered saline (PBS) and then incubated for 30 minutes at 37°C with 50 nM Tubulin Tracker Green (Oregon Green 488 Taxol, bis-Acetate, Life Technologies, Grand Island, NY) in PBS. The samples were then rinsed twice with PBS and re-immersed in PBS solution for imaging. Fluorescence images were acquired using a standard Fluorescein isothiocyanate -FITC filter: excitation of 495 nm and emission 521 nm.

## Cells treated with Taxol and Blebbistatin

For the experiments on chemically modified cells, we have treated the neurons with either: 1) Taxol (10 μM concentration) or 2) Blebbistatin (10 μM concentration), which have been added to the neuron growth medium at the time of plating. Previous work has shown that a concentration of 10 μM of Taxol was very effective in suppressing the microtubule dynamics [10,21,30], and that 10 μM of Blebbistatin was very efficient in disrupting actin polymerization and the formation of actin bundles, thus reducing traction forces between the neurons and the growth substrates [31].

## Data analysis

Growth cone position, axonal length, and angular distributions were measured and quantified using ImageJ (National Institute of Health). The displacement of the growth cone was obtained by measuring the change in the center of the growth cone position. Examples of images that shows the tracked position for axons are shown in S3 Fig. To measure the growth cone velocities the samples were imaged every $\Delta t$ = 5 min for a total period of 1 hr per sample. The 5 min time interval between measurements was chosen such that the typical displacement $\Delta \vec{L}$ of the growth cone in this interval satisfies two requirements: a) is larger than the experimental precision of our measurement (~ 0.1 μm) [19,20]; b) the ratio $\Delta \vec{L}/\Delta t$ accurately approximates the instantaneous velocity $\vec{V}$ of the growth cone. The speed of the growth cone is defined as the magnitude of the velocity vector: $V(t) = |\vec{V}(t)|$, and the growth angle $\alpha(t)$ is measured with respect to the long axis of the semi-cylinder (growth angle and the long axis are defined in Fig 1B). For the data analysis presented in this paper we use the angular coordinate $\theta$ defined as the angle complementary to $\alpha$: $\theta = \frac{\pi}{2} \pm \alpha$ (Figs 1B, 1C and 2D).

Experimental data (Figs 2 and S3) shows that over a distance of ~ 20 μm the axons can be approximated by straight line segments, with a high degree of accuracy. Therefore, to obtain the angular distributions for axonal growth (Fig 3) we have tracked all axons using ImageJ and then partitioned them into segments of 20 μm in length, following the same procedure outlined in our previous work [16,20]. Next, we have recorded the angle $\theta$ that each segment makes with the $x$ direction (defined as the direction perpendicular to the geometrical patterns (Figs 1C and 2D). The total range [0, 2π] of growth angles was divided into 18 intervals of equal size $\Delta\theta_0 = \pi/9$ (Fig 3). To obtain the speed distributions (Fig 4), the range of growth cone speeds at each time point was divided into 15 intervals of equal size $|\Delta \vec{V}_0|$.

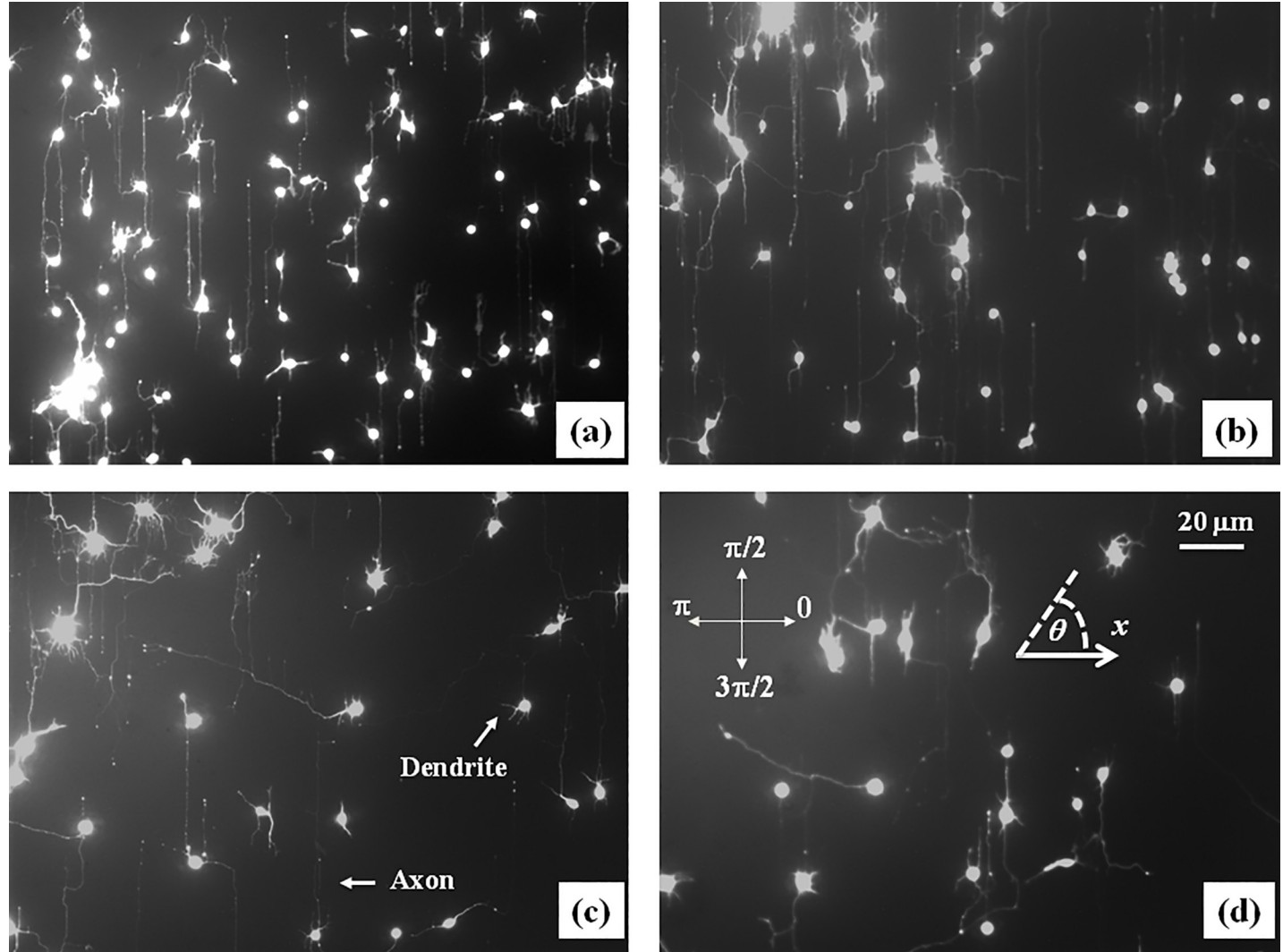

**Fig 2. Examples of fluorescence images of axonal growth for cortical neurons cultured on PDL coated PDMS surfaces with periodic micropatterns.** (*a-b*) Examples of growth for untreated cortical neurons grown on PDMS substrates with pattern spatial period: $d = 4$ µm in (*a*), and $d = 5$ µm in (*b*). (*c*) Examples of axonal growth for cortical neurons treated with Taxol, a chemical compound that inhibits the microtubules dynamics. (*d*) Examples of axonal growth for cortical neurons treated with Blebbistatin, a chemical compound that inhibits the actin dynamics The pattern spatial period is $d = 4$ µm for both (c), (d). The main structural components of a neuronal cell are labeled in (*c*). Cortical neurons typically grow a long process (the axon) and several minor processes (dendrites). The axons is identified by its morphology and the growth cone is identified as the tip of the axon. The angular coordinate used in this paper is defined in (*d*). The directions corresponding to $\theta = 0$, $\pi/2$, $\pi$, and $3\pi/2$ are shown in (*d*). The angles are measured with respect to the $x$ axis, defined as the axis perpendicular to the direction of the PDMS patterns (see Fig 1). All images are captured 42 hrs after neuron plating. The scale bar shown in (*d*) is the same for all images.

### Simulations of growth cone trajectories

We perform simulations of growth cone trajectories using the stochastic Euler method with N steps [26,27]. With this method the change in position of the growth cone and the turning angle at each step are parametrized by the arclength $s$ from the axon's initial position:

$$\Delta x(s) = \cos(\theta) \cdot \Delta s$$

$$\Delta y(s) = \sin(\theta) \cdot \Delta s$$

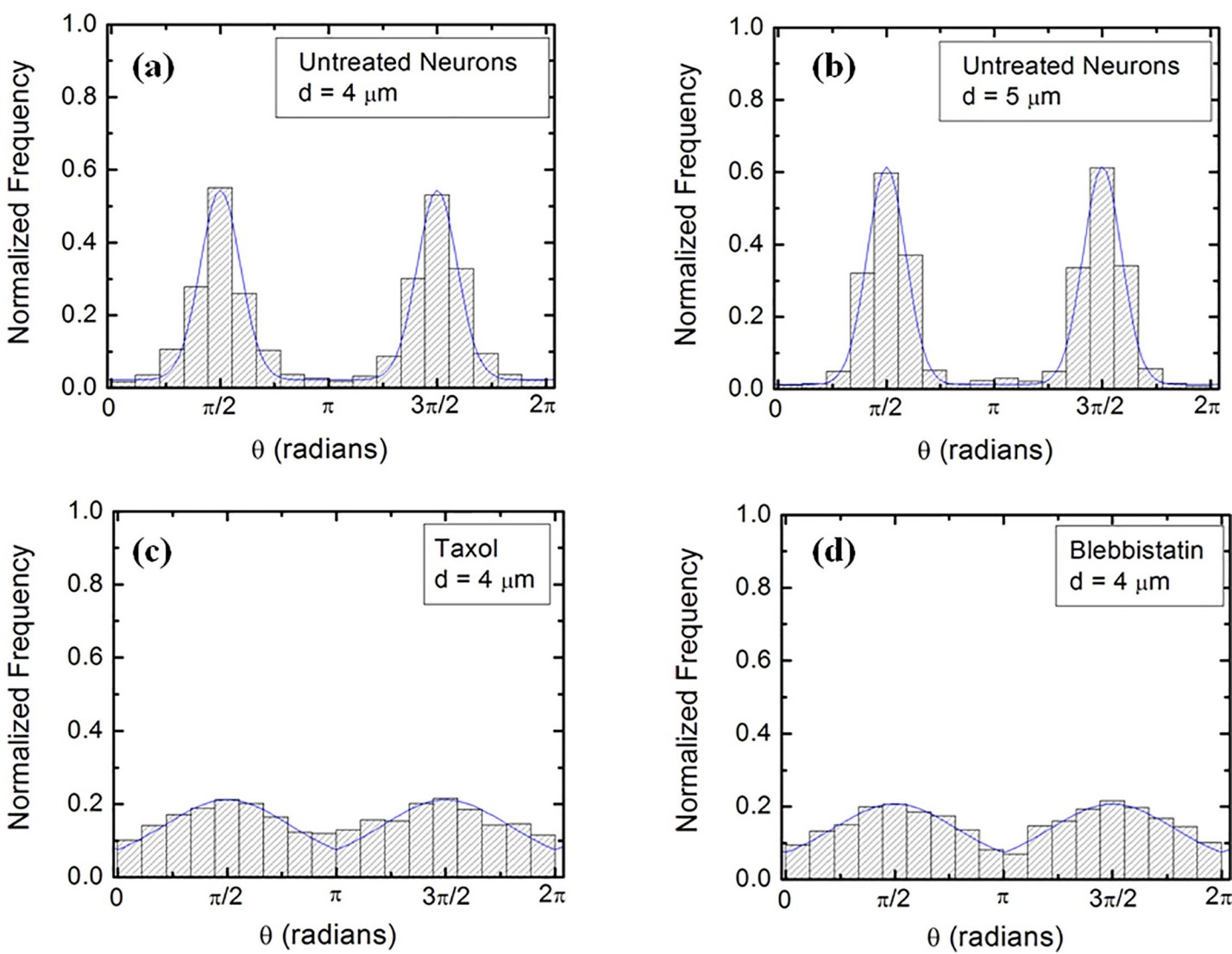

**Fig 3. Examples of normalized experimental angular distributions for axonal growth for neurons cultured on micropatterned PDMS surfaces with different pattern spatial periods _d_.** The vertical axis (labeled Normalized Frequency) represents the ratio between the number of axonal segments growing in a given direction and the total number N of axon segments. Each axonal segment is of 20 μm in length (see section on Data Analysis). All distributions show data collected at _t_ = 42 hrs after neuron plating. The continuous blue curves in each figure are the predictions of the theoretical model discussed in the main text. (_a_) Angular distribution obtained for N = 1440 different axon segments for untreated neurons cultured on surfaces with _d_ = 4 μm (corresponding to Fig 2A). (_b_) Angular distribution obtained for N = 1569 different axon segments for untreated neurons cultured on surfaces with _d_ = 5 μm (corresponding to Fig 2B). The data shows that the axons display strong directional alignment along the surface patterns (peaks at _θ_ = π/2 and _θ_ = 3π/2), with high degree of alignment given by the sharpness of the distributions. (_c_) Angular distribution obtained for N = 1017 different axon segments for neurons treated with Taxol and cultured on surfaces with _d_ = 4 μm (corresponding to Fig 2C). (d) Angular distribution obtained for N = 981 different axon segments for neurons treated with Blebbistatin and cultured on surfaces with on _d_ = 4 μm (corresponding to Fig 2D). The neurons treated with Taxol and Blebbistatin show a significant decrease in the degree of alignment with the surface patterns, compared to the untreated cells.

$$\Delta\theta(s) = -\gamma_\theta \cdot \cos(\theta) + D_\theta \cdot dW$$

where $-\gamma_\theta \cdot \cos(\theta)$ is a deterministic steering torque, and $D_\theta \cdot dW$ is an uncorrelated Wiener process representing the randomness in the axon steering ($\gamma_\theta$ and $D_\theta$ represent the damping and diffusion coefficients, respectively, which are defined in the main text, see Eq (1)). The angle $\theta$

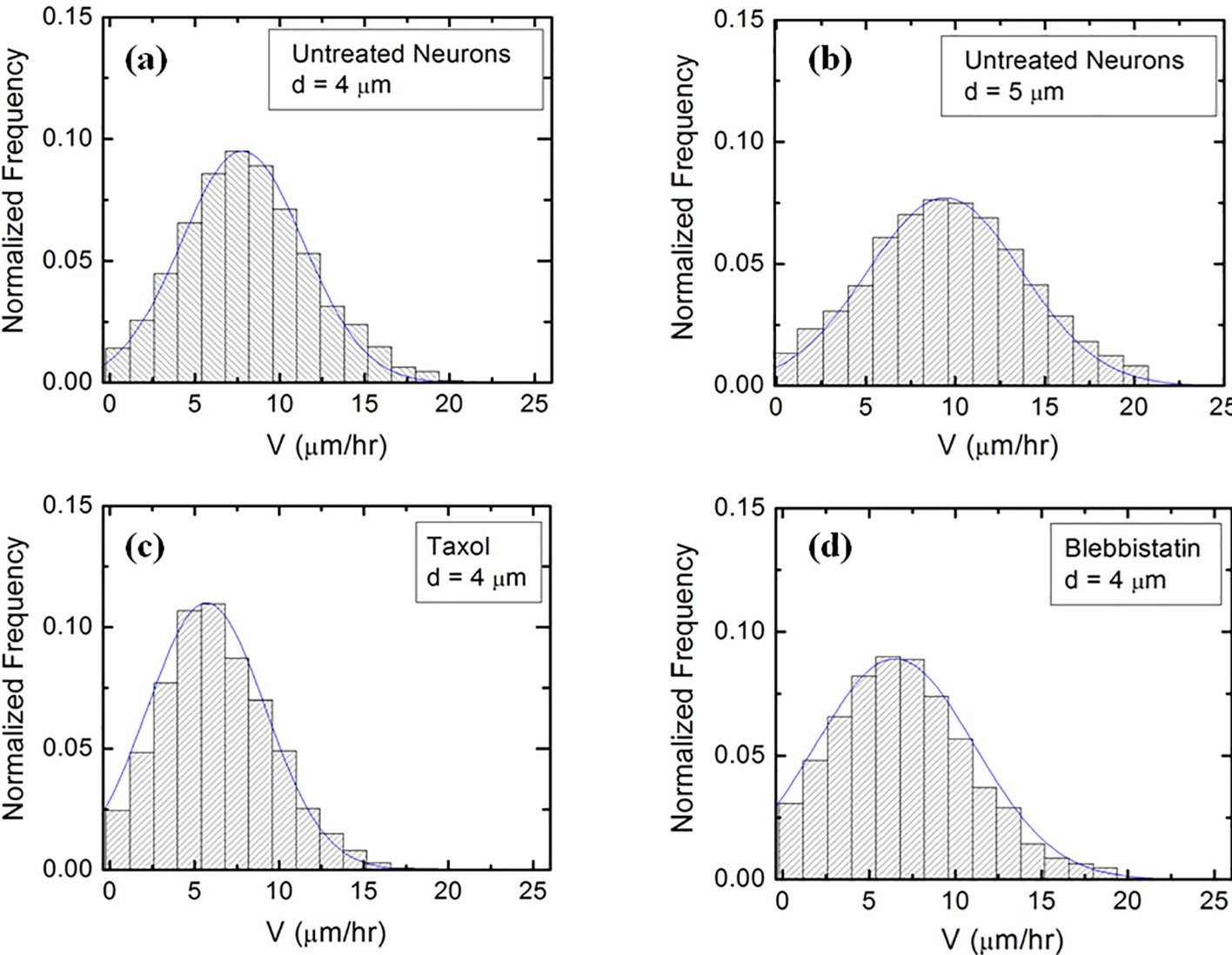

**Fig 4. Examples of normalized speed distributions for growth cones measured on micropatterned PDMS surfaces with different pattern spatial period *d*.** All distributions show data collected at $t = 42$ hrs after neuron plating. The continuous blue curves in each figure are data fits with the theoretical model discussed in the main text. (*a*) Speed distribution for N = 510 different growth cones measured for untreated neurons cultured on surfaces with $d = 4$ μm. (*b*) Speed distribution for N = 526 different growth cones measured for untreated neurons cultured on surfaces with $d = 5$ μm. (*c*) Speed distribution for N = 484 different growth cones measured for Taxol treated neurons cultured on surfaces with $d = 4$ μm. (*d*) Speed distribution for N = 465 different growth cones measured for Blebbistatin treated neurons cultured on surfaces with $d = 4$ μm.

is determined from the angular probability distribution (Eqs (1) and (2) in the main text). The velocity distributions are obtained from the change in position of the growth cone at each step [26,27].

## Results

### Experimental results for axonal growth

Cortical neurons are cultured on PDL coated PDMS surfaces with parallel micropatterns (periodic parallel ridges separated by troughs). The surfaces differ by the value of the pattern spatial period *d* defined as the distance between two neighboring ridges (Fig 1A), and by the radius of curvature *R* of the semi-cylindrical geometrical pattern (Fig 1B). Both parameters were

measured using AFM for each growth surface. We analyze the growth of both untreated and chemically modified neuronal cells on surfaces with spatial periods in the range $d = 1$ μm to 6 μm (in increments of 1 μm), which correspond to curvature radii $R = 0.5$ μm to 1 μm (in increments of 0.1 μm).

Fig 2A and 2B show examples of images of axonal growth for untreated neurons, cultured on PDL coated PDMS micropatterned surfaces with pattern spatial period: $d = 4$ μm (Fig 2A), and $d = 5$ μm (Fig 2B). We have previously demonstrated that axons of untreated neurons display maximum alignment along PDMS patterns for surfaces where the pattern spatial period $d$ matches the linear dimension of the growth cone $l$, where $l$ is in the range 2 to 6 μm [20]. In previous work [16] we demonstrated that the axons tend to grow on top of the semi-cylindrical patterns (as shown schematically in Fig 1B). The experimental data shown in Fig 2A and 2B is in agreement with our previous findings. Examples of the corresponding axonal normalized angular distributions are shown, respectively, in Fig 3A and 3B.

To further investigate the axonal dynamics on PDMS surfaces with periodic micropatterns we measured angular and speed distributions for neurons treated with chemical compounds known to inhibit the dynamics of the cell cytoskeleton. Fig 2C and 2D show examples of axonal growth for neurons treated with 10 μM of Taxol (Fig 2C) and 10 μM of Blebbistain (Fig 2D), and cultured on surfaces with a pattern spatial period $d = 4$ μm. Additional images of axonal growth for neurons treated with Taxol and Blebbistatin and cultured on micropatterned surfaces with $d = 5$ μm are shown in S4A and S4B Fig. The corresponding angular and speed distributions for these axons are shown in S5 and S6 Figs, respectively All images for untreated as well as chemically modified neurons were captured at $t = 42$ hrs after cell plating.

Taxol is a chemical compound which is commonly used to inhibit the normal functioning of the cytoskeleton, due to the disruption of microtubule dynamics [10,21,30]. Blebbistatin is a chemical compound known to inhibit the formation of actin bundles and the reorganization of actin based structures during neuronal growth [31,34]. Both of these compounds were effective at the a concentration of 10 μM used in our experiments [10,21,30,31,34]. The corresponding normalized angular distributions for axonal growth for neurons treated with these chemical compounds are shown in Fig 3C and 3D, respectively. The neurons treated with either Taxol or Blebbistatin show a dramatic decrease in the degree of alignment with the surface patterns compared to the unmodified cells. The data show that while the axonal directionality was greatly reduced by the chemical treatments, the treated neurons still grew long axons and formed cell-cell connections (Figs 2C, 2D, S4A and S4B). These results demonstrate that the disruption of the cytoskeletal dynamics for chemically treated neurons affects only the degree of alignment with the surface pattern, leaving the navigation of the growth cone and axonal outgrowth uninhibited.

Fig 4 shows that the speed distributions for both untreated and chemically modified growth cones are close to Gaussian distributions. This is indeed expected for culture times $t = 42$ hrs after cell plating, as shown in our previous work [16].

## Stochastic model for axonal growth

Axonal dynamics on the PDMS substrates is characterized by both deterministic and stochastic components [10,16,19,20,23]. The angular motion of axons on patterned PDMS surfaces is described by the growth angle $\theta$ defined in Figs 1 and 2D. In our previous work we have shown that the probability distribution $p(\theta,t)$ for the growth angle satisfies the following Fokker-Planck equation [16]:

$$\frac{\partial}{\partial t}p(\theta,t) = \frac{\partial}{\partial \theta}\left[-\gamma_\theta \cdot \cos\theta(t) \cdot p(\theta,t)\right] + D_\theta \cdot \frac{\partial^2}{\partial \theta^2}p(\theta,t) \qquad (1)$$

where $D_\theta$ represents the effective angular diffusion (cell motility) coefficient, and $\gamma_\theta \cdot \cos \theta(t)$ corresponds to a "deterministic torque" representing the tendency of the growth cone to align with the preferred growth direction imposed by the surface geometry [16]. The stationary solution of Eq 1 is given by [16]:

$$p(\theta) = A \cdot \exp\left(\frac{\gamma_\theta}{D_\theta} \cdot |\sin(\theta)|\right) \qquad (2)$$

where $A$ is a normalization constant obtained from the normalization condition: $\int_0^{2\pi} p(\theta) \cdot d\theta = 1$.

The absolute value $|\sin \theta|$ in Eq 2 reflects the symmetry of the growth around the $x$ axis: the angular distributions centered at $\theta = \pi/2$ and $\theta = 3\pi/2$ are symmetric with respect to the directions $\theta = \pi$ and $\theta = 0$ (Fig 3). This is a consequence of the fact that there is no preferred direction along the PDMS pattern (Figs 1 and 2), and this feature applies to all types of micropatterned PDMS surfaces, and for both untreated and chemically modified neurons. We also note that the deterministic torque has a maximum value if the growth cone moves perpendicular to the surface patterns ($\theta = 0$ or $\theta = \pi$), in which case the cell-surface interaction tend to align the axon with the surface pattern. The torque is zero for an axon moving along the micropattern.

The speed distribution $p(V,t)$ of axonal growth is given by [16]:

$$\frac{\partial}{\partial V} p(V,t) = \frac{\partial}{\partial V}[\gamma_s \cdot (V - V_s) \cdot p(V,t)] + \frac{\sigma^2}{2} \cdot \frac{\partial^2}{\partial \theta^2} p(V,t) \qquad (3)$$

where $\gamma_s$ is the constant damping coefficient of the corresponding Langevin equation ($\gamma_s = 1/\tau$ where $\tau$ is the characteristic decay time), $V_s$ is the average stationary speed of the neuron population, and $\sigma$ is noise strength for an uncorrelated Wiener process with Gaussian white noise [16,28,29]. Eq 3 has the following stationary solution [16]:

$$p(V) = C \cdot \exp\left(-\frac{\gamma_s}{\sigma^2} \cdot (V - V_s)^2\right) \qquad (4)$$

where $C$ is a normalization constant obtained from the normalization condition: $\int_0^\infty p(V) \cdot dV = 1$.

Eqs 1–4 define a purely kinematic model that describes the axonal motion using stochastic terms. This model predicts that the overall motion for the axons has two components: a) a uniform drift along the directions of the PDMS micropattern (*i.e.*, along the long axis of the semicylinder), and b) a random walk around these equilibrium positions. This is indeed what is observed experimentally. At small times the growth cone dynamics resembles a Brownian motion, resulting in a slow increase in the mean growth cone position along the $x$ axis. As time progresses the axon exhibits a feedback control (see below) which steers the axonal motion along the micropatterned parallel PDMS lines (Figs 1 and 2). Furthermore, in the absence of the micropatterns the motion for the growth cones reduces to regular diffusion (Ornstein—Uhlenbeck) process characterized by exponential decay of the autocorrelation functions with a characteristic time $\tau = \frac{1}{\gamma_\theta}$, axonal mean square length that increases linearly with time, and velocity distributions that approach Gaussian functions [28,29]. In our previous work we have shown that this was indeed the case for axonal growth on PDL coated glass or for PDMS surfaces characterized by large pattern spatial periods: $d > 9$ μm [16,20].

We used Eqs 2 and 4 to fit the normalized experimental angular and speed distributions for each type of surface and cell (untreated or chemically modified) considered in our experiments (fits to the data are represented by the continuous blue curves in Figs 3 and 4). For the case of

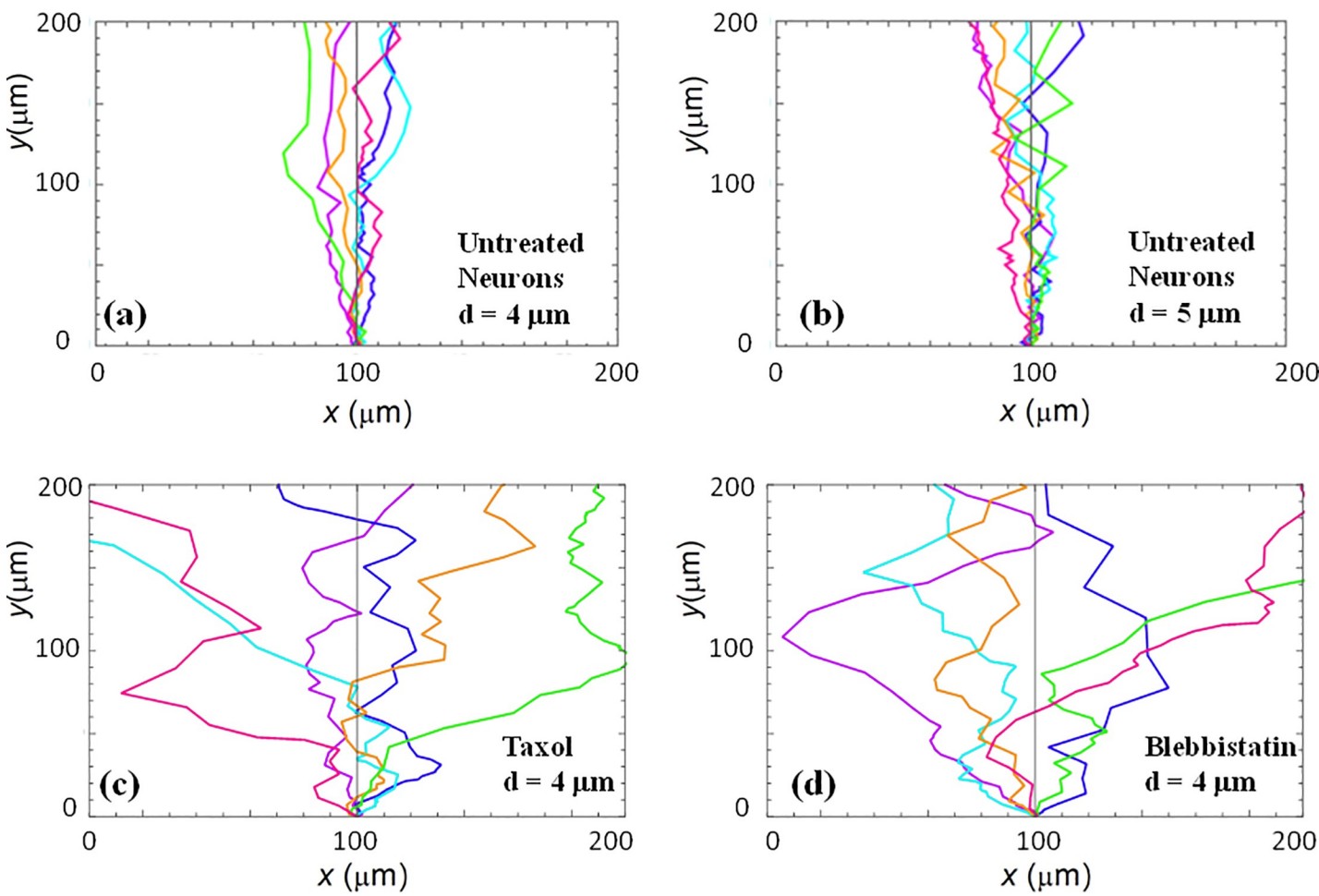

**Fig 5. Examples of simulated axonal growth.** (*a-b*) Untreated neurons. (*c*) Neurons treated with Taxol. (*d*) Neurons treated with Blebbistatin. The simulations are performed by using the values of the growth parameters obtained from the fit of the experimental data with Eqs 2 and 4 (seen main text). The pattern spatial periods correspond to the data shown in Figs 3 and 4: *d* = 4 μm for (a), (c), (d), and *d* = 5 μm for (b) respectively.

untreated neurons, the theoretical model fit the experimental data for the angular probability distributions with the values for the ratio between the deterministic torque and the diffusion coefficient for the angular motion in the range: $\gamma_\theta/D_\theta \approx 2$ to 9. These values are consistent with the values reported in our previous work [10,16,20] for similar growth times (t = 42 hrs after plating). Similarly, for neurons treated with Taxol we obtain from the data fit: $\gamma_\theta/D_\theta \approx 1$ to 4.5, while for the neurons treated with Blebbistatin we get: $\gamma_\theta/D_\theta \approx 1$ to 3.5. A detailed examination of these results is presented in the Discussion section below. Here we note that the values for the ratio of the growth parameters decrease upon the chemical treatment of the neurons. In S1 Table in the Supporting Information we give a summary of the values for the guiding parameter $\gamma_\theta/D_\theta$ compared to the corresponding values measured for various surfaces in our previous work [10,16,20].

We used the solutions for the probability distributions given by the theoretical model presented above to simulate axonal growth trajectories, as well as axonal speed and angular distributions. The simulations were performed using the values for ratios between the deterministic torque and the angular diffusion coefficient obtained from the fit to the experimental data (Figs 3 and 4) with *no additional* adjustable parameters. Fig 5 shows examples of simulation

results for untreated (Fig 5A and 5B and Taxol-treated neurons grown on surfaces with $d = 4$ μm (Fig 5C and 5D). Additional simulations results performed in the case of chemically modified neurons cultured on surfaces with $d = 5$ μm are shown in S7 Fig.

We emphasize that the angular distributions and speed distributions obtained from these simulations match the experimental data for untreated, Taxol, and Blebbistatin treated neurons *without* the introduction of any *additional* parameters. For example, the simulated axon trajectories in Fig 5A and 5B reproduce the high degree of alignment observed experimentally for untreated neurons grown on surfaces with $d = 4$ μm or $d = 5$ μm (Figs 2A, 2B, 3A and 3B). Figs 5C, 5D, S7A and S7B. show simulated growth trajectories with an intermediate degree of alignment (similar to the data measured on Taxol and Blebbistatin treated neurons in Figs 2C and 2D, 3C and 3D, as well as S4 and S5 Figs).

## Mechanical model for axonal dynamics

We justify the kinematic model described by Eqs 1–4 by introducing a simple mechanical model that takes into account the cell-substrate interactions. An earlier version of this model was first proposed in reference [35]. The model considers the bending-induced strained sustained by the axon while growing on the semi-cylindrical surface (Fig 1B): axonal adhesion to the surface leads to axonal bending, which in turn leads to increased mechanical strain energy in the axon cytoskeleton. The mechanical strain energy $E$ depends on the axon bending stiffness $B$, and the local surface curvature $K(\theta,R)$ [35]:

$$E = \frac{1}{2} B \cdot K^2(\theta, R) \tag{5}$$

In the case of axonal growth on the micropatterned surfaces, the curvature of the axon on the surface of the semi-cylinder is given by:

$$K^2(\theta, R) = \frac{|\cos(\alpha)|}{R^2} \equiv \frac{|\sin(\theta)|}{R^2} \tag{6}$$

Under the maximum entropy assumption, the probability of axon growing in a given direction is given by Boltzmann distribution:

$$p(\theta) = A_1 \cdot \exp\left(-\frac{E}{E_0}\right) \tag{7}$$

where $E_0$ is the characteristic energy scale for axonal bending, and $A_1$ is an overall normalization constant. By combining Eqs 5–7, and using our convention for the angular variable $\theta$ (Figs 1B, 1C and 2D) we can write the following expression for the angular probability distribution:

$$p(\theta) = A \cdot \exp\left(\frac{B}{E_0 \cdot R^2} \cdot |\sin(\theta)|\right) \tag{8}$$

A direct comparison between Eq 8 (derived from the mechanical beam model) and Eq 2 (solution of the stochastic Fokker-Planck equation) leads to the following relationship between the stochastic parameters (deterministic torque $\gamma_\theta$ and angular diffusion coefficient $D_\theta$) and the mechanical parameters (axon bending stiffness $B$, the characteristic energy scale $E_0$, and

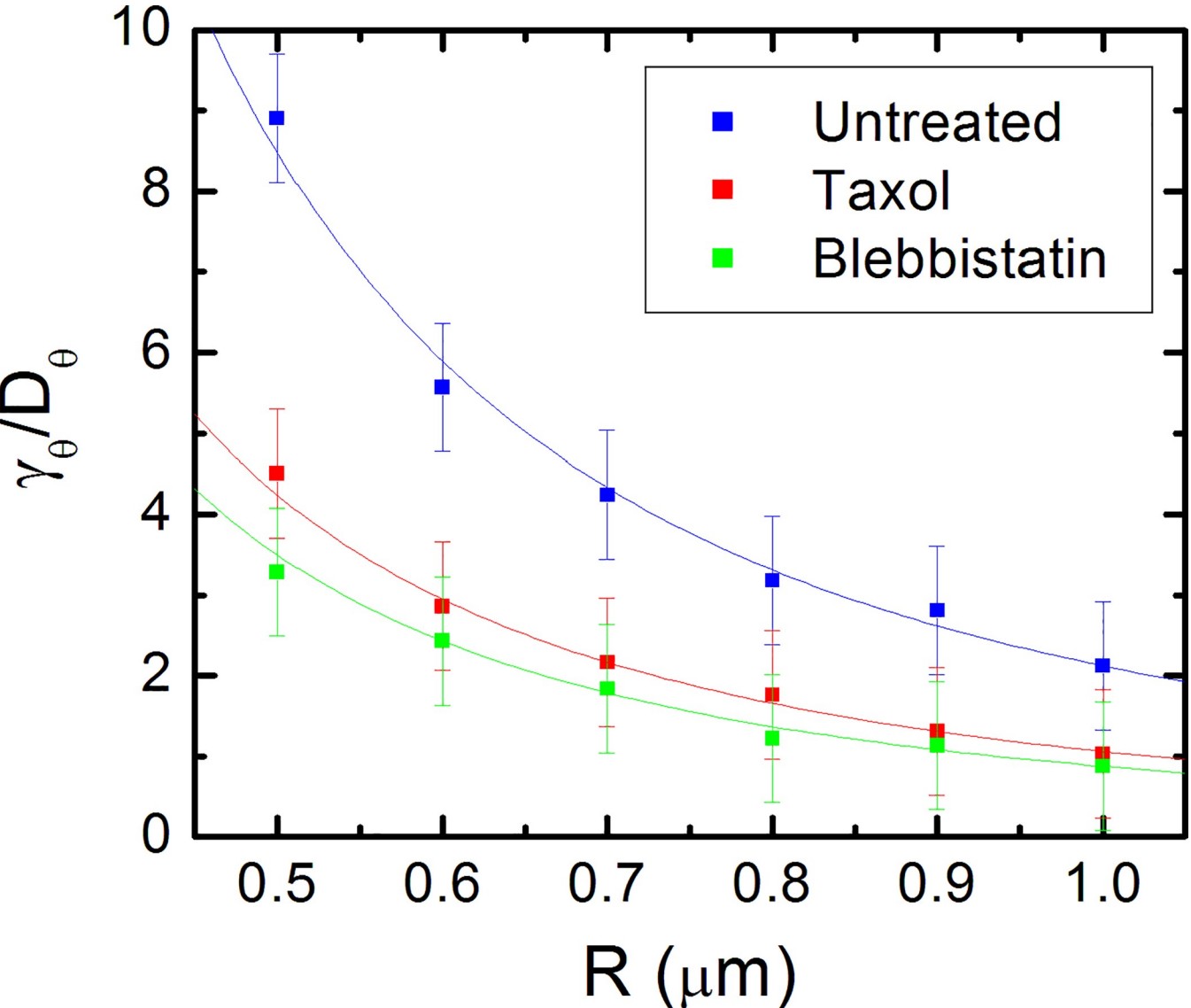

**Fig 6. Variation of the ratio $\gamma_\theta/D_\theta$ with the pattern radius of curvature $R$ for axonal growth on micropatterned PDMS substrates.** The guiding parameter $\gamma_\theta/D_\theta$ represents the ratio between the deterministic torque and the angular diffusion coefficient (control parameters), and the radius of curvature of the semi-cylindrical pattern $R$ represents the external geometrical stimulus. The blue squares represent the values for $\gamma_\theta/D_\theta$ obtained from the fit to experimental data for untreated neurons. The red squares correspond to the experimental data obtained for neurons treated with Taxol, while the green squares correspond to the data measured for neurons treated with Blebbistatin. Error bars indicate the standard error of the mean for each data set. The dotted curves represent fit of the data points with Eq 9.

the surface radius of curvature $R$):

$$\frac{\gamma_\theta}{D_\theta} = \frac{B}{E_0 \cdot R^2} \tag{9}$$

To further investigate this model, we measure the variation of the ratio between the deterministic torque and the angular diffusion coefficient $\gamma_\theta/D_\theta$ (obtained from the fit of the experimental angular distributions as described in the previous section) with the radius of curvature of the semi-cylindrical pattern $R$ measured by AFM.

Fig 6 shows the variation of the experimentally measured values for $\gamma_\theta/D_\theta$ with $R$, for untreated neurons (blue squares), as well as for neurons treated with Taxol (red squares) and Blebbistatin (green squares). The continuous curves in Fig 6 represent fits to the data with Eq 9. The data points in Fig 6 are fitted by the mechanical model for a constant value of the characteristic energy scale $E_0 = (3.5\pm0.9)\cdot10^{-15}$ J and the following values for the axonal bending rigidity: $B = (0.59\pm0.07)$ nN·μm$^3$ for untreated neurons, $B = (0.31\pm0.05)$ nN·μm$^3$ for neurons treated with Taxol, and $B = (0.20\pm0.07)$ nN·μm$^3$ for neurons treated with Blebbistatin. These values for the axonal bending rigidity are comparable to the values for axon bending rigidity reported in the literature [35,36], as well as with our previous values for the elastic modulus of the neuron, if we assume a simple rigid beam model for the axon [30–32].

Fig 6 together with the simple mechanical model and the simulations presented in the previous section imply that axonal motion on surfaces with periodic geometries exhibits a closed-loop feedback behavior: the growth cone detects the geometrical cues on the surface and tends to align its motion along certain preferred directions that maximize the cell-surface interactions. In general, feedback control means that the system is steered towards a target behavior by using information which is retrieved from the environment through continuous measurements. This is a powerful technique for describing the dynamical properties of many types of physical and biological systems including particle trapping [37–39], optical tweezers [40–42], neuron firing [43,44] and cellular dynamics [45–47].

## Discussion

Neurons respond to a variety of external cues (biochemical, mechanical, geometrical) while wiring up the nervous system *in vivo* [1,2,4–7]. In many cases these cues consist in periodic geometrical patterns with dimensions of a 0.1 to 5 microns [2,4,5]. These physiological growth scaffolds include curved brain folding [4–6], radial glial fibers [4,5], and extracellular matrix tracks [1,2,4–7]. In addition to these *in vivo* cues, the directional guidance of neurons has been extensively examined *in vitro* on a variety of growth surfaces that contain micron—size features such as grooves, steps or ridges [3,8–18]. Thus bioengineered substrates with periodic geometrical features could be used as nerve implants and bridging scaffolds to direct regeneration of severed nerve tracts in the nervous system [1–9]. Our studies show that growth substrates containing micropatterned periodic features promote axonal growth along the direction of the pattern. The range for the micropattern spatial periods in our experiments ($d$ = 1 to 6 μm) is relevant both for neuronal growth *in vivo* as well as for many proposed biomimetic implants for nerve regeneration [9,14]. Our experiments show that neurons grown on micropatterned PDMS substrates display a significant (factor of 5–10) increase in the overall axonal length compared to growth on un-patterned (PDMS or glass) surfaces. We also demonstrate a high degree of alignment when the pattern spatial period $d$ matches the linear dimension of the growth cone: $d \approx l$. Furthermore, the radius of curvature $R$ of the geometrical patterns matches the typical curvature found on neuron scaffolds (extracellular matrices) *in vivo*.

Fundamentally, axonal growth on surfaces with controlled geometries arises as the result of an interplay between deterministic and stochastic components of growth cone motility [10,16,19,20,23]. An example of a deterministic influence is the presence of a preferred direction of growth along a specific geometric pattern on a substrate [10,16]. Examples of stochastic influences are the effects of polymerization of cytoskeletal features such as actin filaments and microtubules, cell signaling, low concentration biomolecule detection, biochemical reactions within the neuron, and the formation of lamellipodia and filopodia [1–9,23–25]. The resultant growth cannot be predicted for individual neurons due to this stochastic-deterministic

interplay, however the defining features of a population of neurons can be modeled by probability functions that satisfy a set of well-defined stochastic differential equations [10,16,19–23]. Foremost examples include the Langevin and Fokker-Planck equations, which have been used previously to capture the effects of both deterministic and stochastic influences on cellular motion, which is treated as a form of biased random motion [10,15,16,19–27]. Models based on these equations have been used to successfully characterize many aspects of the probabilistic behavior of cell populations, including angular and speed correlation functions, distributions of velocity and acceleration, mean square displacements, as well as diffusion (cell motility) coefficients [16,19–27].

In this paper we demonstrate that the dynamics of the growth cones on surfaces with micropatterned periodic features is described by stochastic Fokker-Planck equations. The stochastic growth is based on a simple mechanical model that takes into account the interaction between axon and the growth substrates and captures all the main characteristics of axonal growth for untreated and chemically modified neurons: deterministic torque and diffusion (cell motility) coefficients, angular and speed distributions, and axon bending rigidity. Furthermore, this model implies a simple closed-loop feedback model for axonal motion: the growth cone detects geometrical features on the substrate and orients its motion in the directions that maximize the interaction between the axon and the substrate. Models based on the theory of feedback control systems have been successfully used by other groups to characterize the galvanotaxis (motion in external electric fields) of human granulocytes and keratinocytes [22,45], as well as the chemotactic response of bacteria and of various types of virus modified cells [46,47].

Our results show that the closed-loop feedback control underlies the mechanism of axonal alignment on micropatterned PDMS substrates when the pattern spatial period $d$ matches the linear dimension of the growth cone: $d \approx l$. Within this model the growth cone behaves similarly to a "device" that senses geometrical cues, and as a result generates traction forces that align the axon with the surface pattern. This behavior is displayed by both untreated and chemically modified neurons, as shown in Fig 6. In this figure each data set (for untreated, Taxol and Blebbistatin treated cells) is fitted with a unique parameter which demonstrates that the axonal response is proportional to the signal received from the guidance cue [45,46]. The data in Fig 6 also demonstrates that the response of the feedback control system is affected by the inhibition of cytoskeletal dynamics: the actual response (measured by the deterministic torque and the bending stiffness) is different for the untreated and chemically treated cells. In all these cases the pattern spatial period $d$ and the pattern radius of curvature $R$ play the role of an effective external (geometric) stimulus that determines the axonal alignment, similar to the electric field in the case of galvanotaxis of human granulocytes and keratinocytes [22,45], or the concentration gradient in the case of cellular chemotaxis [46].

These results support our previous findings that neurons follow geometrical patterns through a contact–guidance mechanism [10,20]. Contact guidance is the ability of cells to change their motion in response to geometrical cues present in the surrounding environment. This behavior has been observed for several types of cells including neurons, fibroblasts, and tumor cells [10,14,20,48]. Previous work [14,48–50] has shown that growth cones develop several different types of curvature sensing proteins (such as amphipathic helices and bin-amphiphysin-rvs (BAR) —domains) that act as sensors of geometrical cues and are involved in the generation of traction forces. Moreover, the degree of directional alignment of cellular motion is increasing with the increase in the density of curvature sensing proteins [49,50]. In our experiments the growth cone filopodia and lamellipodia wrap around the ridges of the PDMS micropatterns [16], which results in a minimal contact area with the surface, and thus a maximum density of curvature sensing proteins. Consequently, high-curvature geometrical features such as ridges on PDMS substrates will impart higher forces to the focal contacts of filopodia wrapped over these features, compared to

the low-curvature patterns. This means that the contact guidance mechanism leads to an increase in the traction force along the direction of the surface pattern, which ultimately results in the observed directional alignment of axons on these surfaces.

Growth cones are filled up with actin filaments, which polymerize at their leading edge [1–6]. At the same time, myosin II motors pull on actin filaments and generate traction forces via point contacts (integrins, viniculin, talin etc.). Furthermore, interactions between actin filaments and microtubules, modify the distribution of mechanical stress in the growth cone and affect its adhesion properties, and its ability to navigate and turn. In consequence, both microtubules and actin filaments inside the growth cone act as stiff load-bearing structures that generate surface adhesion and traction forces [1,2,4]. Inhibition of microtubule or actin dynamics will therefore result in a decrease in cell-substrate interactions and adhesion. Our experiments demonstrate that disruption of the cytoskeletal dynamics for cells treated with Taxol (inhibitor of microtubule dynamics), and Blebbistatin (disruption of actin filaments) results in a decrease in the degree of alignment and a reduction in cell-substrate interactions (Figs 2C, 2D, 3C, 3D, S4 and S5). Furthermore, the smaller values of the guiding parameter $\gamma_\theta/D_\theta$ and bending rigidity $B$ for the chemically treated neurons implies a less effective guidance mechanism and cell-substrate mechanical coupling for these cells compared to the untreated ones. Thus, the results obtained in the case of chemically treated neurons show an alteration of the feedback control system responsible for directional motion of axons. These experimental results are in agreement with the predictions of the contact guidance mechanisms discussed above.

We have shown that the Fokker-Planck equation provides a general stochastic framework that describes the main characteristic of the axonal dynamics on micropatterned PDMS substrates. The limitations of this model are due to its phenomenological nature: the growth parameters are obtained from fit to experimental data, and not predicted from the underlying cellular biophysics. However, we have demonstrated that a simple mechanical model based on the axon bending-induced strain justifies the use of the Fokker-Planck equation and allows us to extract the main dynamical parameters: guiding parameter and bending rigidity for both untreated and chemically modified neurons. Our results show that the additional cues necessary to guide the axonal dynamics result from the interplay between the geometrical features of the substrate and the physical properties (stiffness) of the nerve process.

The theoretical model presented in this paper could be further extended to account for the explicit dependence of the growth parameters on the mechanical and biochemical guidance cues, such as changes in the stiffness of the growth substrate, change in adhesion properties, or external chemical gradients. This approach could provide significant insight into the neuronal response to external stimuli, without the need to incorporate all the molecular steps involved. The model discussed here could also be applied to other types of cells to give new insight into the nature of cellular motility. In future experiments we propose to directly measure cell-surface coupling forces using traction force microscopy as well as the density of cell surface receptors and curvature sensing proteins using high-resolution fluorescence techniques. In principle these future investigations will enable researchers to quantify the influence of environmental cues (geometrical, mechanical, biochemical) on cellular dynamics, and to relate the observed cell motility behavior to cellular processes, such as cytoskeletal dynamics, cell-surface interactions, and signal transduction.

## Conclusions

In this paper, we have performed a detailed experimental and theoretical analysis of axonal growth on micropatterned PDMS surfaces. We have demonstrated that the axonal dynamics on these surfaces is described by a theoretical model based on the motion of a closed-loop feedback system on a substrate with periodic geometrical features. We have used this model to

measure the growth parameters that characterize the axonal motion. Our results show that axonal dynamics is regulated by a contact-guidance mechanism, which stems from cell-surface mechanical interactions and cellular feedback in an external periodic geometry: the axon responds to geometrical cues by rotating and aligning its motion along the surface micropatterns. The general model presented here could be applied to describe the dynamics of other types of cells in different environments including external electric fields, substrates with various mechanical properties, and biomolecular cues with different concentration gradients.

## Supporting information

**S1 Fig. Variation of the surface roughness on micropatterned PDMS surfaces with spatial periods *d*.** The surface roughness is measured with the AFM. The green data points show the average surface roughness measured before PDL coating. The blue data points show the average surface roughness measured on the same surfaces after coating with PDL. The error bars indicate the standard error of the mean. The data demonstrates that the surface roughness does not vary significantly among the PDMS surfaces with different spatial periods *d*, and it does not change significantly upon surface coating with PDL. The variation of the average roughness among these substrates is less than 10%.
(TIF)

**S2 Fig. Examples of histograms for the substrate elastic modulus *E* for micropatterned PDMS surfaces measured with the AFM.** (a) Histogram for elastic modulus for a PDMS surface with d = 4 μm, measured before coating the surface with PDL. (b) Histogram for elastic modulus for the same PDMS surface shown in (a), measured after coating the surface with PDL. The two maps display similar ranges for *E*. The average elastic modulus between the two maps differs by only 5%. The data demonstrates that PDL coating does not change the elastic modulus of the PDMS substrate.
(TIF)

**S3 Fig. Examples of tracked positions for axons.** The segments marked in yellow are superimposed on the axons and show the growth trajectory. The numbers on each segment represent different positions of the growth cone during growth. Each segment is 20 μm in length as described in the Data Analysis section in the main text.
(TIF)

**S4 Fig.** Fluorescence images showing examples of axonal growth for cortical neurons treated with Taxol (a) and Blebbistatin (b). The images are captured 42 hrs after neuron plating. The scale bar shown in (a) is the same for both images. The pattern spatial period is *d* = 5 μm for both images.
(TIF)

**S5 Fig.** Normalized experimental angular distributions for axonal growth measured for neurons treated with Taxol (a) or Blebbistatin (b). The angular distributions are measured for cell cultured on micropatterned PDMS surfaces with pattern spatial period is *d* = 5 μm. The vertical axis (labeled Normalized Frequency) represents the ratio between the number of axonal segments growing in a given direction and the total number N of axon segments. Each axonal segment is of 20 μm in length (see section on Data Analysis in the main text). All distributions show data collected at *t* = 42 hrs after neuron plating. (a) Angular distribution obtained for N = 734 different axon segments for neurons treated with Taxol. (b) Angular distribution obtained for N = 694 different axon segments for neurons treated with Blebbistatin. The neurons treated with either Taxol or Blebbistatin show a significant decrease in the degree of

alignment with the surface patterns, compared to the untreated cells. The continuous blue curves in each figure are the predictions of the theoretical model presented in the main text.
(TIF)

**S6 Fig.** Normalized speed distributions obtained for growth cones of cortical neurons treated with Taxol (a) and Blebbistatin (b). The growth substrates are PDL coated PDMS surfaces with periodic micropatterns with the pattern spatial period $d = 5$ μm. (a) Speed distribution measured for N = 221 different growth cones for neurons treated with Taxol. (b) Speed distribution measured for N = 206 different growth cones for neurons treated with Blebbistatin. The continuous blue curves in each figure are the predictions of the theoretical model presented in the main text.
(TIF)

**S7 Fig.** Examples of simulated neuronal growth for neurons treated with Taxol (a) or Blebbistatin (b). The simulations are performed by using the values of the growth parameters obtained from the fit of the experimental data with Eqs 2 and 4 (seen main text). The pattern spatial period for both images is $d = 5$ μm, corresponding to the experimental data shown in S4–S6 Figs.
(TIF)

**S1 Table. Values for the guiding parameter $\gamma_\theta/D_\theta$.** The guiding parameter decreases with increases pattern curvature radius $R$, as well as upon chemically treatment of neurons. All measurements are performed at the same time (t = 42 hrs after plating).
(PDF)

## Author Contributions

**Conceptualization:** Cristian Staii.

**Data curation:** Cristian Staii.

**Formal analysis:** Jacob P. Sunnerberg, Marc Descoteaux, Cristian Staii.

**Funding acquisition:** Cristian Staii.

**Investigation:** Cristian Staii.

**Methodology:** David L. Kaplan, Cristian Staii.

**Project administration:** Cristian Staii.

**Resources:** Cristian Staii.

**Supervision:** Cristian Staii.

**Validation:** Cristian Staii.

**Writing – original draft:** David L. Kaplan, Cristian Staii.

**Writing – review & editing:** David L. Kaplan, Cristian Staii.

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
