## [Decision Letter · Decision Letter 0]

7 Jul 2021

PONE-D-21-19160

Axonal growth on surfaces with periodic geometrical patterns

PLOS ONE

Dear Dr. Staii,

Thank you for submitting your manuscript to PLOS ONE. After careful consideration, we feel that it has merit but does not fully meet PLOS ONE’s publication criteria as it currently stands. Therefore, we invite you to submit a revised version of the manuscript that addresses all the points raised during the review process. The reviewers are very positive and ask for discussions or precisions on a fex points. Please address these concerns carefully.

We look forward to receiving your revised manuscript.

Kind regards,

Etienne Dague, PhD

Academic Editor

PLOS ONE

Journal Requirements:

"The authors gratefully acknowledge financial support for this work from Tufts Springboard (CS,DLK), National Science Foundation award DMR 2104294 (CS), and Tufts Physics Summer Program (MD)."

"CS, National Science Foundation award DMR 2104294"

Reviewers' comments:

Reviewer's Responses to Questions

**Comments to the Author**

1. Is the manuscript technically sound, and do the data support the conclusions?

Reviewer #1: Yes

Reviewer #2: Partly

2. Has the statistical analysis been performed appropriately and rigorously? 

Reviewer #1: Yes

Reviewer #2: I Don't Know

3. Have the authors made all data underlying the findings in their manuscript fully available?

Reviewer #1: No

Reviewer #2: Yes

4. Is the manuscript presented in an intelligible fashion and written in standard English?

Reviewer #1: Yes

Reviewer #2: Yes

5. Review Comments to the Author

Reviewer #1: The authors develop a model of axonal motility which incorporates mechanical interactions between the axon and the growth substrate. They have published in this domain with different substrates in the past. Hence a scaling argument based on fokker-planck parameters would be great to have for the comparison of current study and previously published work by this group and others. Also-why is the cell density is low? is there any way to compare their density, neuron length, or any other relevant features of neurons with representative tissues found in mammals?

Reviewer #2: This is an interesting article that follows up on a myriad of data showing biochemical, mechanical and geometric wiring of the nervous system.

One main concern is that the article is based on mechanical and adhesive properties but the drugs used to block the growth are actin/microtubule inhibitors. Somewhere this needs to be clarified and why adhesive properties were not modulated to test the hypothesis for growth cone extension

There needs to be, at the very least, a discussion regarding the choice of the model. Why would the Markov process or other algorithms fit/not fit? What is the limitation of the Fokker-Plank equation.

Did you note a finite growth, experimentally or theoretically? There needs to be a bit of a discussion here on how bioengineering substrates would allow for growth in the (long or short) distance?

Could you elaborate a bit more on how the d=1-6um is relevant for invivo and biomimetic implant (Line 410) ?

6. PLOS authors have the option to publish the peer review history of their article (what does this mean?). If published, this will include your full peer review and any attached files.

Reviewer #1: No

Reviewer #2: No

---

## [Author Response · Author response to Decision Letter 0]

6 Aug 2021

Please see the Response to Reviewers file.

Based on the reviewer’s comments we have revised the manuscript as described in detail below. 

Reviewer #1

The authors develop a model of axonal motility which incorporates mechanical interactions between the axon and the growth substrate. They have published in this domain with different substrates in the past. Hence a scaling argument based on fokker-planck parameters would be great to have for the comparison of current study and previously published work by this group and others. Also-why is the cell density is low? is there any way to compare their density, neuron length, or any other relevant features of neurons with representative tissues found in mammals?

We thank the referee for their overall assessment. Our responses to the referee’s questions and amendments to the paper are presented below. All changes are highlighted in yellow in the “Revised Manuscript with Track Changes” document. 

Regarding the scaling argument, we first note that in the paper we compare the ratio of the growth parameters (deterministic torque/diffusion coefficient i.e. \\gamma_\\theta/D_\\theta ) for untreated neurons vs. chemically modified neurons (last paragraph on page 15). This ratio is obtained from the Fokker-Planck model and it is referred to as the guiding parameter in the paper. We also plot the variation of guiding parameter with the radius of curvature of the pattern (Fig. 6). The guiding parameter is a measure of the effectiveness of the guiding mechanism as we discuss on the first paragraph on page 23, and it is decreasing upon the chemical treatment of the neurons. In the revised paper we added a table (S1 Table in the Supporting Information) in which we compare the values of the guiding parameter obtained in this work with the corresponding values obtained in our previous work for untreated neurons grown for various surfaces (flat PDMS, glass, directional polymer surfaces). All these different measurements are performed at the same time (t = 42 hrs after plating). We refer to this table in the first paragraph on page 16. 

We note that Fokker-Planck equation is a general stochastic equation that describes axon dynamics, as well as the dynamics of many different types of cells (please see for e.g. references 22-27). In this paper we show that a simple mechanical model based on axon-bending induced strain justifies the use of Fokker-Planck equation and allows us to extract the main dynamical parameters for axonal motion. We emphasize that this model: a) fully accounts for the experimental data of neuronal growth on micropatterned PDMS surfaces, including growth speeds, axonal alignment, terminal velocities, and angular distributions; 2) has a minimum number of phenomenological parameters that account for the cell-surface interactions; and 3) allows for meaningful comparisons between untreated and chemically modified neurons. However, this phenomenological model does not give a simple scaling relation that contains all chemical modifications of neurons, all time scales or all growth substrates. A full description of neuron dynamics at different time scales and for different types of surfaces will likely have to take into account the explicit dependence of axonal growth on the cell-surface interactions. This will lead to a refined model that would require a detailed investigation of the interactions between the axons and the growth substrate, using traction force microscopy and fluorescence techniques. We propose to perform these detailed investigations in future work, as discussed in the first paragraph on page 24 in the revised manuscript. In the revised manuscript we have included several paragraphs to clarify these important points: middle paragraph on page 20 and the last paragraph on page 23.

Regarding cell density and axonal length, the values reported in this paper are relevant for neurons in the early stages of the development of the nervous system in mammals. In previous work (references 10, 15, 16 and 20) we have found experimentally that neuron densities in the range 3000 – 7000 cells/cm2 are optimal for studying the effect of geometrical cues on neuronal growth on various types of surfaces. For example, in reference 15 we show that neurons do not grow long processes at low culture density (smaller than 2000 cells/ cm2). As the cell density increases, the degree of axonal alignment decreases, which reflects the fact that the axons are making more connections at higher densities, therefore deviating from the direction imparted by the surface geometry. This implies that high surface densities (higher than 8,000 cells/ cm2) where neuron-neuron signaling is important are also sub-optimal for exploring the effect of physical cues on neuronal growth. The cell density of 5000 cells/cm2 chosen in this paper is in the middle of this optimal density range. This is also close to the neuron density in vivo in the early stages of development of the nervous system (see references 1, 2, 4). A direct consequence of this relatively low cell density is that in the theoretical model we take the neuron-neuron signaling to be negligible compared to neuron-substrate interactions.

In the revised manuscript we have included several paragraphs to address these points: last paragraph in the Cell Culture section on page 8, and first paragraph in the Discussion section (pages 19 and 20). 

Reviewer #2

This is an interesting article that follows up on a myriad of data showing biochemical, mechanical and geometric wiring of the nervous system.

We thank the reviewer for their overall positive assessment. Our responses to the reviewer’s questions and amendments to the paper are presented below. All changes are highlighted in yellow in the “Revised Manuscript with Track Changes” document. 

One main concern is that the article is based on mechanical and adhesive properties but the drugs used to block the growth are actin/microtubule inhibitors. Somewhere this needs to be clarified and why adhesive properties were not modulated to test the hypothesis for growth cone extension.

The experiments presented in this paper demonstrate the importance of the cytoskeleton and molecular motors (myosin II) in controlling the axonal alignment on directional substrates. Previous experiments (references 48-50) have shown that the development of curvature sensing proteins and focal adhesion points depends on external forces, including cell-substrate traction forces. Myosin II controls the dynamics of actin filaments at the leading edge of the growth cone, and thus plays an essential role in the generation of external forces, and the maturation of curvature sensing proteins and focal adhesion points. Furthermore, interactions between actin filaments and microtubules, modify the distribution of mechanical stress in the growth cone and affect its adhesion properties, and its ability to navigate and turn. Inhibition of actin or microtubule dynamics will therefore result in a decrease in cell-substrate interactions and adhesion. Blebbistatin (Taxol) are chemical compounds that inhibits the activity of myosin II (microtubules), and thus alters the dynamics of actin filaments. Consequently, this affects the generation of traction forces, the development of curvature sensing proteins, and ultimately leads to a decrease degree of alignment of the axons with the surface patterns. Our experiments show that while the axonal directionality is greatly reduced by the chemical treatment, the treated neurons are still growing axons and form multiple cell-cell connections (Fig. 2 c,d). These results demonstrate that only the degree of alignment with the surface pattern is affected by the disruption of cytoskeletal dynamics, while the navigation of the growth cone is not changed significantly. This enables us to perform a direct comparison between untreated neurons and chemically modify cells, and to develop a theoretical model that accounts for the experimental data, as discussed in the paper.

In future studies we propose to perform a detailed investigation of the surface adhesion properties and of the interactions between the growth cones and the growth substrates. We propose to perform these measurements by using a combination of traction force and fluorescence microscopy (first paragraph on page 24). While the elucidation of the biophysical and biochemical underpinnings of the observed dynamical behavior requires future measurements, the experimental results and the simplified model presented here provide a framework for investigating axonal alignment and contact guidance and could be very useful for designing novel substrates to guide the regeneration of severed nerve tracts. This approach could be further extended and applied to other types of cells, to account for the explicit dependence of the cell motility parameters on the environmental cues, such as chemical gradients, electric fields, or substrate stiffness.

In the revised manuscript we have added several paragraphs (last paragraph on page 22, the first and last paragraph on page 23) to address these important points. 

There needs to be, at the very least, a discussion regarding the choice of the model. Why would the Markov process or other algorithms fit/not fit? What is the limitation of the Fokker-Plank equation.

Fokker-Planck equation is a general stochastic equation that describes axon dynamics (as well as the dynamics of many different types of cells). Models based on Fokker-Planck equation, Brownian motion, or more generally Markov processes are extensively used in literature to describe the motility of many different types of cells, including neuronal growth (please see for e.g. references 22-27, as well as our previous work references 10, 16, 20). The usefulness of these models stems from the fact that, fundamentally, cellular motion arises as the result of an interplay between deterministic and stochastic components that affect cell motility. For axons, an example of a deterministic influence is the presence of a preferred direction of growth along a specific geometric pattern on a substrate. Examples of stochastic influences are the effects of polymerization of cytoskeletal elements such as actin filaments and microtubules, cell signaling, low concentration biomolecule detection, biochemical reactions within the neuron, and the formation of lamellipodia and filopodia. In the case of neurons, the resultant growth cannot be predicted for individual axons due to this stochastic-deterministic interplay, however the defining features of a population of neurons can be modeled by probability functions that satisfy the Fokker-Planck equation. 

In this paper we show that a simple mechanical model based on axon-bending induced strain justifies the use of Fokker-Planck equation and allows us to extract the main dynamical parameters for axonal motion. We emphasize that this model: a) fully accounts for the experimental data of neuronal growth on micropatterned PDMS surfaces, including growth speeds, axonal alignment, terminal velocities, and angular distributions; 2) has a minimum number of phenomenological parameters that account for the cell-surface interactions; and 3) allows for meaningful comparisons between untreated and chemically modified neurons. We note that a full description of neuron dynamics at different time scales and for different types of surfaces will likely have to take into account the explicit dependence of axonal growth on the cell-surface interactions. This will lead to a refined model that would require a detailed investigation of the interactions between the axons and the growth substrate, using traction force microscopy and fluorescence techniques. We propose to perform these detailed investigations in future work as discussed in the first paragraph on page 24 in the revised manuscript. 

In the revised manuscript we have included several paragraphs (middle paragraph on page 20 and the last paragraph on page 23) to clarify these points. The limitations of this method are discussed in the last paragraph on page 23. 

Did you note a finite growth, experimentally or theoretically? There needs to be a bit of a discussion here on how bioengineering substrates would allow for growth in the (long or short) distance?

Axons grow until they make connections with dendrites from other neurons. We note that with respect to cell density and axonal length, the values reported in this paper are relevant for neurons in the early stages of the development of the nervous system in mammals. In previous work (references 10, 15, 16 and 20) we have found experimentally that neuron densities in the range 3000 – 7000 cells/cm2 are optimal for studying the effect of geometrical cues on neuronal growth on various types of surfaces. For example, in reference 15 we show that neurons do not grow long processes at low culture density (smaller than 2000 cells/ cm2). As the cell density increases, the degree of axonal alignment decreases, which reflects the fact that the axons are making more connections at higher densities, therefore deviating from the direction imparted by the surface geometry. This implies that high surface densities (higher than 8,000 cells/ cm2) where neuron-neuron signaling is important are also sub-optimal for exploring the effect of physical cues on neuronal growth. The cell density of 5000 cells/cm2 chosen in this paper is in the middle of this optimal density range. Moreover, our experiments show that neurons grown on micropatterned PDMS substrates display a significant (factor 5-10) increase in the overall axonal length compared to un-patterned PDMS or glass surfaces. In the revised manuscript we have included several paragraphs to address this issue: last paragraph in the Cell Culture section on page 8, and the last paragraph in the Discussion section (pages 19 and 20). 

Could you elaborate a bit more on how the d=1-6um is relevant for invivo and biomimetic implant (Line 410) ?

Our results show contact guidance mechanism for axonal growth, which implies that the axonal orientation is affected by the topography of the substrate, as discussed on page 22. For the in vivo case, numerous studies have shown that micron-size geometrical features act as physiological growth scaffolds, including curved brain folding, radial glial fibers, and extracellular matrix tracks. However, neuronal growth has not being investigated quantitatively in most of these literature reports. In addition to these in vivo cues, the directional guidance of neurons has been extensively examined in vitro on a variety of growth surfaces that contain micron - size features such as scratches, grooves, steps or ridges. Thus bioengineered substrates with periodic geometrical features could be used as nerve implants and bridging scaffolds to direct regeneration of severed nerve tracts in the nervous system. In the revised manuscript we have included a paragraph (first paragraph on page 19 in the Discussion section) to address this point.

---

## [Decision Letter · Decision Letter 1]

8 Sep 2021

Axonal growth on surfaces with periodic geometrical patterns

PONE-D-21-19160R1

Dear Dr. Staii,

We’re pleased to inform you that your manuscript has been judged scientifically suitable for publication and will be formally accepted for publication once it meets all outstanding technical requirements.

Kind regards,

Etienne Dague, PhD

Academic Editor

PLOS ONE

Additional Editor Comments (optional):

Reviewers' comments:

Reviewer's Responses to Questions

**Comments to the Author**

1. If the authors have adequately addressed your comments raised in a previous round of review and you feel that this manuscript is now acceptable for publication, you may indicate that here to bypass the “Comments to the Author” section, enter your conflict of interest statement in the “Confidential to Editor” section, and submit your "Accept" recommendation.

Reviewer #1: All comments have been addressed

Reviewer #2: All comments have been addressed

2. Is the manuscript technically sound, and do the data support the conclusions?

Reviewer #1: Yes

Reviewer #2: Yes

3. Has the statistical analysis been performed appropriately and rigorously? 

Reviewer #1: Yes

Reviewer #2: I Don't Know

4. Have the authors made all data underlying the findings in their manuscript fully available?

Reviewer #1: Yes

Reviewer #2: Yes

5. Is the manuscript presented in an intelligible fashion and written in standard English?

Reviewer #1: Yes

Reviewer #2: Yes

6. Review Comments to the Author

Reviewer #1: all comments have been addressed by the authors; i recommend publication in its current format without changes.

Reviewer #2: (No Response)

7. PLOS authors have the option to publish the peer review history of their article (what does this mean?). If published, this will include your full peer review and any attached files.

Reviewer #1: No

Reviewer #2: No

---

## [Editor Report · Acceptance letter]

14 Sep 2021

PONE-D-21-19160R1 

Axonal Growth on Surfaces with Periodic Geometrical Patterns 

Dear Dr. Staii:

I'm pleased to inform you that your manuscript has been deemed suitable for publication in PLOS ONE. Congratulations! Your manuscript is now with our production department. 

Kind regards, 

on behalf of

Dr. Etienne Dague 

Academic Editor

PLOS ONE